# Impact of Sarcopenia and Inflammation on Patients with Advanced Non-Small Cell Lung Cancer (NCSCL) Treated with Immune Checkpoint Inhibitors (ICIs): A Prospective Study

**DOI:** 10.3390/cancers13246355

**Published:** 2021-12-17

**Authors:** Marta Tenuta, Alain Gelibter, Carla Pandozzi, Grazia Sirgiovanni, Federica Campolo, Mary Anna Venneri, Salvatore Caponnetto, Enrico Cortesi, Paolo Marchetti, Andrea M. Isidori, Emilia Sbardella

**Affiliations:** 1Department of Experimental Medicine, Sapienza University of Rome, Viale Regina Elena 324, 00161 Rome, Italy; marta.tenuta@uniroma1.it (M.T.); carla.pandozzi@uniroma1.it (C.P.); federica.campolo@uniroma1.it (F.C.); maryanna.venneri@uniroma1.it (M.A.V.); andrea.isidori@uniroma1.it (A.M.I.); 2Medical Oncology Unit B, Policlinico Umberto I, Sapienza University of Rome, 00185 Rome, Italy; alain.gelibter@uniroma1.it (A.G.); grazia.sirgiovanni@uniroma1.it (G.S.); salvo.caponnetto@uniroma1.it (S.C.); enrico.cortesi@uniroma1.it (E.C.); paolo.marchetti@uniroma1.it (P.M.)

**Keywords:** sarcopenia, lung cancer, immunotherapy, PDL1, biomarker

## Abstract

**Simple Summary:**

The association between sarcopenia and therapeutic response in cancer patients treated with immune checkpoint inhibitors (ICI) is still limited to a few retrospective reports. In a population of advanced non-small cell lung cancer (NCSCL) patient candidates for ICI, we demonstrated that sarcopenic status is associated with worse progression free survival and that sarcopenic patients had an 8-fold higher risk of progression disease. Sarcopenia could therefore be considered an independent unfavorable prognostic factor and predictive of a worse response to ICI. The choice to use the DXA scan, in our opinion, allows a better standardization of results than the CT scan. Moreover, patients with sarcopenia showed significantly higher inflammatory markers compared to non-sarcopenic patients. We therefore believed that sarcopenia might be a useful predictive marker for ICI therapy outcomes and that may help identify patients more likely to achieve a better response to ICI in routine clinical practice.

**Abstract:**

Background: Sarcopenia is a condition characterized by loss of skeletal muscle mass associated with worse clinical outcomes in cancer patients. Data on sarcopenia in patients undergoing immune checkpoint inhibitors (ICI) therapy are still limited. The aim of this prospective observational study was to investigate the relationship between sarcopenia, ICI treatment response and immunological profile, in patients with advanced non-small cell lung cancer (NSCLC). Methods: Forty-seven stage IV NSCLC patient candidates for starting ICI, were enrolled from the Policlinico Umberto I outpatient Oncology. Patients underwent baseline blood test, inflammatory markers, cytokine assessment and body composition with dual-energy X-ray absorptiometry (DXA). Sarcopenia was defined with appendicular skeletal muscle mass over height^2^ (ASM/heigh^2^). Results: Overall, 19/47 patients (40.4%) results were sarcopenic. Sarcopenic patients showed significantly shorter PFS than non-sarcopenic ones (20.3 weeks, 95% CI 7.5–33.1 vs. 61 weeks, 95% CI 22.5–99.4, *p* = 0.047). Specifically, they had an 8.1 times higher risk of progression disease (PD) than non-sarcopenic patients (OR 8.1, 95%, *p* = 0.011). Conclusions: Sarcopenic patients showed worse PFS and had a higher risk of PD compared to non-sarcopenic ones. Therefore, sarcopenia may reflect the increased metabolic activity of more aggressive tumors, which involves systemic inflammation and muscle wasting and could be considered a negative predictive factor for ICI response.

## 1. Introduction

Immune checkpoint inhibitors (ICI) represent a new effective weapon in the fight against cancer; in recent years, they have changed the history of many tumors, including non-small cell lung cancer (NSCLC), by enormously increasing patient survival [1,2].

Sarcopenia is a condition characterized by loss of skeletal muscle mass and decreased muscle function, mainly caused by aging, but even malnutrition, chronic diseases and cancer [3]. A tumor itself can also cause significant decreases in muscle mass and be responsible of secondary sarcopenia [4,5]. Sarcopenia occurs in approximately 50% of advanced cancer patients, being related to malnutrition, inflammation and tumor treatments [6]. In these patients, it has been demonstrated to be associated with worse outcomes in terms of overall survival (OS) and progression free survival (PFS), and it can be considered as a negative prognostic factor for different types of cancer, regardless of stage or treatment approach [7,8,9,10].

Moreover, several studies have demonstrated high levels of inflammatory markers and/or pro-inflammatory cytokines in sarcopenic patients, thus suggesting that the inflammatory substrate may have a significant role in the development of sarcopenia itself, as high levels of inflammatory cytokines and acute phase reactant could initiate various molecular pathways in skeletal muscle. Neutrophil/lymphocyte ratio (NLR), tumor necrosis factor alpha (TNF-α), interleukin 6 (IL-6), C-reactive protein (CRP) are the markers most frequently associated with sarcopenic status and they could be considered as risk factors for the development of sarcopenia itself [11,12,13,14,15].

A pro-inflammatory status is also usually found in patients with worse prognosis: in details high levels of leukocyte/lymphocyte ratio (LLR), NLR, absolute count of monocytes, eosinophils, LDH and CRP are usually predictors of worse prognosis [16,17,18,19].

Bearing all this in mind, sarcopenia and systemic inflammation are both associated with poor prognosis, thus supporting the idea that sarcopenia can reflect the increased metabolic activity of more aggressive tumors, which involves systemic inflammation and muscle wasting, as recently demonstrated in a study on head-neck cancer patients [20]. Therefore, sarcopenia, in association with specific inflammatory parameters, seem to have a negative role on the therapeutic response in advanced neoplastic patients.

Sarcopenia is frequently diagnosed in NSCLC patients with worse survival outcomes [21]. However, in the new scenario of ICI, clinical data regarding the association between sarcopenia and ICI efficacy is limited to few retrospective reports [6,22,23,24,25]. A recent meta-analysis on 740 patients with advanced cancer receiving ICI concluded sarcopenia to be an independent, unfavourable prognostic factor in patients with advanced cancer in treatment with ICI [26]. In all the studies considered, it is evaluated through CT scan. This technique is the gold standard for the assessment of muscle mass, even if mostly for retrospective studies, data could be less standardized than Dual-energy X-ray absorptiometry (DXA) [27].

Due to the lack of prospective data in the literature on the association between sarcopenia, treatment with ICI, response to treatment and immunological profile, we designed a prospective observational study in NSCLC patients candidates to initiate treatment with ICI. The primary outcome was to confirm trough DXA scan if sarcopenic status is associated with—or could be predictive of—worse clinical outcomes in patients with advanced NSCLC receiving anti-PD-1/PD-L1. Secondary endpoints included to evaluate the association between sarcopenia and inflammatory status and also the relationship with the immunological profile.

## 2. Materials and Methods

### 2.1. Study Population

The study population included patients entering Oncology Unit B, Policlinico Umberto I of Rome, from October 2017 to February 2020, with diagnosis of advanced NSCLC, candidate to start anti-PD-1 (Nivolumab, 240 mg flat dose every two weeks and Pembrolizumab, 200 mg flat dose, every three weeks) or anti-PD-L1 (Atezolizumab, 1200 mg flat dose every three weeks). Exclusion criteria to participate the study were previous oncologic diagnosis and treatment, pregnancy, previous organ transplantation and physical disabilities. The evaluation of the patients’ clinical response was carried out until 30 November 2021.

This study was approved by the Ethics Committee of Policlinico Umberto I (Rif. CE 4946) and was conducted in accordance with the Declaration of Helsinki principles. Each patient provided a written informed consent to participate the study.

### 2.2. Study Design and Procedures

This is a proof of concept, prospective longitudinal observational study and part of a wider study on endocrine adverse events of ICI on patients with NSCLC in treatment with PD-1/PD-L1 inhibitors.

Patients were evaluated prior to start ICI therapy. Clinical medical history was recorded and anthropometric measures were assessed (weight, height and BMI). Patients underwent a blood test to evaluate routine full blood chemical tests (full blood count, coagulation, lipid and glucose metabolism), inflammatory markers (erythrocyte sedimentation rate, ESR; CRP; fibrinogen; ferritin; transferrin) and cytokine assessment (IL-6, TNF-α, transforming growth factor alpha, TGF-α).

Body composition was assessed with DXA, Hologic Inc., Marlborough, MA, USA, QDR 4500 W and was measured for the whole body and in specific body regions. Delimiters for regional analysis were determined by standard software (Hologic Inc., Marlborough, MA, USA, S/N 47168 VER. 11.2). Coefficient of variation was <1.5% for body fat. The diagnosis of sarcopenia, according to the updated European Working Group on Sarcopenia in Older People (EWGSOP2) [28], was based on low muscle quantity: appendicular skeletal muscle mass (ASM; kg) was calculated as the sum of the upper and lower limbs lean mass. ASM was adjusted for body size by dividing ASM over height squared (ASM/heigh^2^, kg/m^2^). Cut-offs value to define sarcopenia were ASM/heigh^2^ of <7.0 kg/m^2^ in men and < 5.5 kg/m^2^ in women [28]. Therefore, a stratification of the cohort was performed based on the presence of sarcopenia or not.

Response to treatment was assessed in accordance with the Response Evaluation Criteria in Solid Tumors (RECIST) version 1.1. Patients were categorized into two main groups based on best response: (1) clinical benefit group (CB), including complete (CR), partial response (PR) or stable disease (SD); (2) progression disease group (PD), including patients with disease progression.

### 2.3. Statistical Analyses

Outcome measurements were assessed for normality using the Shapiro–Wilk test, and non-parametric tests were used when violations of parametric test assumptions were evident. Values were then expressed as median and interquartile range (IQR). The Mann–Whitney U test was used to determine whether there were differences between covariates in the different cohorts.

PFS was calculated from the date of initiation of anti-PD-1/anti-PD-L1 therapy to the date of disease progression or death and was censored at the date of the last visit for patients who were still alive without any documented disease progression. OS was calculated from the date of initiation of anti-PD-1/anti-PD-L1 therapy to the date of death or date of last visit for patients with no confirmation of death.

Kaplan–Meier survival analysis was conducted to assess the median time to PFS and OS for sarcopenic and non-sarcopenic groups, and pairwise log rank comparisons were conducted to determine which group had significantly different survival distributions. A binomial logistic regression was performed to ascertain the effects of sarcopenia on the likelihood of developing PD, corrected appropriately.

The *p*-values were two sided for all statistical tests and *p* < 0.05 was considered to be statistically significant. All statistical analyses were performed with SPSS Statistics version 27.0 (IBM SPSS Statistics Inc., Chicago, IL, USA).

## 3. Results

### 3.1. Characteristics of the Cohort

Between October 2017 and February 2020, a total of 60 patients were screened to enter the study. However, 5 of them were excluded for sudden death (*n* = 1) or transfer to other hospitals (*n* = 4). Finally, 8 patients could not perform DXA scan due to clinical conditions.

Overall, the final cohort consisted of 47 patients, 27 males (57.4%) and 20 females (42.6%). Median age of the population was 67 years (61; 74). All patients had NSCLC, distributed as follows based on histological type: 30 adenocarcinoma (63.8%), 9 squamous cell carcinoma (19.1%), 5 poorly differentiated carcinoma (10.6%), 3 large cell or mixed carcinoma (6.4%). Regarding treatment type: 22 patients (46.8%) received Nivolumab: 240 mg flat dose every 15 days (Opdivo^®^, Bristol–Myers Squibb, New York, NY, USA) as a second or third line treatment; 18 patients (38.3%) Pembrolizumab: 200 mg flat dose every 21 days (Keytruda^®^, Merck Sharp & Dohme Corp., Kenilworth, NJ, USA) as a first or second line treatment; 7 patients (14.9%) Atezolizumab: 1200 mg flat dose every 21 days (Tecentriq^®^, Roche S.p.A, Milan, Italy) as second or third line treatment. In detail, a total of 17 patients (36.2%) started anti-PD-1/PD-L1 as first-line therapy, 22 patients (46.8%) as second line and 8 (17.0%) as third line therapy.

If we compare first line treatment (*n* = 17) with patients who used anti-PD-1 as second and third line (*n* = 30), no differences were found in PD at chi-square test (7/18 vs. 10/29; *p* = 0.760). 

General characteristics of patients in baseline are shown in Table 1.

Based on the best response, 63.8% of patients (*n* = 30) were included in the CB group (*n* = 12 PR or CR; *n* = 18 SD) while 36.2% (*n* = 17) were included in the PD group. During the study period, 44 patients (93.6%) had to interrupt treatment with ICI due to PD or toxicity. Overall, 36 patients (76.6%) died during the observation period. The median PFS was 30.3 weeks (13.0; 73.1) while the median OS was 65.6 weeks (27.0; 113.3). Overall, the response rate (ORR) was 25.5% (Table 1).

With regard to body weight only one patient fell into the underweight category (BMI < 18.5 kg/m^2^), 26 patients were of normal weight (BMI 18.5–24.9 kg/m^2^) while 20 patients were overweight or obese (BMI > 25 kg/m^2^) (Table 2).

### 3.2. Primary Outcome: Sarcopenia and Clinical Response

Overall, 19/47 patients (40.4%) were identified to have sarcopenia according to ASM/heigh^2^. No gender differences were found in prevalence of sarcopenia: it was described in 10/23 males (52.6%) and 9/19 females (47.4%) (*p* = 0.582). BMI was lower in patients with sarcopenia: 20.9 (20; 26.8) vs. 25.6 (22.4; 30.4), *p* = 0.008. However, taking into account BMI subgroups, the only underweight patient was not sarcopenic. Among normo-weight patients 53.8% (*n* = 14/26) and 25% (*n* = 5/20) among overweight/obese patients were sarcopenic. No differences in regard to BMI were seen both in normo-weight and overweight/obese patients (Table 2).

There were no significant differences in histology, smoking status, number of prior therapies between patients with and without sarcopenia.

Characteristics of metabolic and body composition parameters are summarized in Table 2. Specifically, a significant higher number of patients had sarcopenia in the PD group compared to CB group (11/17, 64.7% vs. 8/30, 36.4%, *p* = 0.011). Indeed, ASM/heigh^2^ (*p* = 0.014) and lean mass (*p* = 0.05) were lower in PD group than in CB group (Table 2).

Patients with sarcopenia showed shorter PFS than those without sarcopenia (20.3 weeks, 95% CI 7.5–33.1 vs. 61 weeks, 95% CI 22.5–99.4, *p* = 0.047). No differences instead were found in OS (49.4 weeks, 95% CI 39.8–59.0 vs. 80.4 weeks, 95% CI 52.8–108.1, *p* = 0.304) (Figure 1, panels A,B). ORR was lower in sarcopenic patients (10% vs. 35.7%, *p* = 0.05).

A subgroup analysis for patients who performed ICI as first line (*n* = 17) and patients who performed therapy as second and third line (*n* = 30) was performed for survival curves. Results show that for first line treatment, both PFS and OS were shorter in patients with sarcopenia: PFS 3.0 weeks, 95% CI 0–21.0 vs. 82.6 weeks, 95% CI 0–170.1, *p* = 0.013; OS 23.3 weeks, 95% CI 0–68.7 vs. 88.1 weeks, 95% CI 28.5–147.7, *p* = 0.030 (Figure 1, Panels C,D). Conversely, no differences were found between sarcopenic and non-sarcopenic patients both for PFS and OS in patients who underwent ICI as second and third line treatment: PFS 25.0 weeks, 95% CI 0–62.57 vs. 43.4 weeks, 95% CI 2.0–84.8, *p* = 0.531; OS 87.1 weeks, 95% CI 19.9–154.3 vs. 71.0 weeks, 95% CI 56.7–85.3, *p* = 0.903 (Figure 1, Panels E,F).

A binomial logistic regression was performed to ascertain the effects of sarcopenia on the likelihood to develop PD as best response to ICI treatment, corrected for age, previous chemotherapy (Table 3). The logistic regression model was statistically significant: χ^2^ (8) = 31.868, *p* < 0.001. The model explained 48.28% (Nagelkerke’s R2) of the variance in PD. Of the four predictor variables, only sarcopenia was statistically significant. Patients with sarcopenia had an 8.1 times higher risk of PD than non-sarcopenic ones (OR 8.1, 95% CI: 1.6–40.9, *p* = 0.011).

### 3.3. Secondary Outcome: Sarcopenia and Inflammatory Biomarkers

Considering full blood count, patients with sarcopenia showed higher white blood cells (*p* = 0.041) and, in details, higher neutrophils (*p* = 0.022). NLR ratio (*p* = 0.012) and LLR ratio (*p* = 0.005) were also higher in sarcopenic patients. Moreover, the percentage of patients with basal neutrophils >4500/μL was significantly higher in sarcopenic patients compared to non-sarcopenic ones (18/19: 94.7% vs. 19/28: 67.8%, *p* = 0.04), while no differences were found considering lymphocytes <1000/μL (*p* = 0.175).

In regard to inflammatory biomarkers, CRP (*p* = 0.024) and fibrinogen (*p* = 0.041) were found to be higher in sarcopenic patients compared to non-sarcopenic ones (Table 4). Even ferritin value was close to statistical significance (*p* = 0.052).

Cytokines also showed a pro-inflammatory pattern in sarcopenic patients: indeed, IL-6 (*p* = 0.004) and TGF-α (*p* = 0.042) were higher in sarcopenic population.

## 4. Discussion

Since their introduction into cancer therapy, ICI are increasingly used in the management of numerous advanced/metastatic cancers.

To the best of our knowledge, this is the first prospective study assessing sarcopenia through DXA scan on a population of advanced NSCLC patients prior to receiving PD-1/PD-L1 inhibitors. We demonstrated that sarcopenic status in these patients is associated with worse survival outcomes in terms of PFS.

Sarcopenia is frequently diagnosed in NSCLC patients [21]. In our cohort, the prevalence of sarcopenia was 40.4%, in line with previous studies on patients with lung cancer [6,29]. BMI does not reflect the sarcopenic status as it represents an imprecise measure of body composition and does not distinguish between fat and lean body mass [30]. In our population, in fact, even if sarcopenic patients had lower BMI than non-sarcopenic patients, weight values were however within the normal range. The only underweight patient was not sarcopenic. Moreover, 53.8% among normo-weight patients and 25% among overweight/obese patients were sarcopenic. The evaluation of BMI itself is therefore not sufficient to characterize the cancer patient for a possible sarcopenic state. On the contrary, it can confuse the clinical picture.

So far, assessing the sarcopenic condition in oncologic patients may be very important. In fact, the association between sarcopenia and worse survival outcomes has already been widely explored in the literature [31]. According to a meta-analysis conducted by Shlomit et al. on 7843 cancer patients, sarcopenia could be considered a negative prognostic factor for different types of tumors, at different stages, in relation to OS [7].

However, in the new scenario of ICI, only a few retrospective studies highlighted how basal sarcopenia is associated with worse prognosis in advanced cancer patients [6,22,23,24,25,32,33,34]. According to a recent systematic review and meta-analysis (nine cohort studies consisting of 740 patients with advanced cancer receiving ICI), sarcopenia resulted as an independent, unfavorable prognostic factor in these patients. Authors concluded that a routine assessment of sarcopenia status and its early correction should be encouraged in order to ensure better outcomes [26]. Authors, however, acknowledge among limitations that data are pooled from retrospective cohort studies, which might have affected the quality of the meta-analysis. They also included studies with different types of advanced cancer, which may have caused significant heterogeneities.

In regard to NSCLC patients treated with ICI, literature data are limited to few recent retrospective studies. According to Nishioka et al. baseline sarcopenia status was associated with worse ORR and PFS in advanced NSCLC treated with ICI [24]. These results were confirmed by other studies who performed regression analysis and concluded that sarcopenia could be considered as an independent predictor of worse outcomes [6,22,23]. Only one report shows no significant differences between sarcopenic and non-sarcopenic group, despite median PFS and OS appear decidedly longer among patients with non-low skeletal muscle mass compared to those with low skeletal muscle mass [25]. In line with literature data, our results showed that sarcopenia is more frequently found in patients who incurred PD as a best response. ORR is also lower in sarcopenic patients. As a matter of fact, sarcopenic patients had a shorter PFS than non-sarcopenic ones. Moreover, based on logistic regression results, sarcopenic patients had an 8-fold higher risk of PD than non-sarcopenic patients, confirming that sarcopenia could be considered an independent unfavorable prognostic factor, and probably predictive of a not brilliant response to ICI.

Unfortunately, these results were not confirmed in term of OS. The explanation of different results between PFS and OS probably lies in the fact that sarcopenia has to be considered as a risk factor for a worse prognosis, but it is not the only one. In fact, survival, especially in this type of patients, can be conditioned by various factors, particularly when the observation time increases.

In this regard, it may be interesting to consider data relating to patients who underwent first-line therapy and, therefore, in which the possible confounding factors are reduced. As a result of the subgroup analysis for patients who performed ICI as first line and patients who performed therapy as second and third line, both PFS and OS were shorter in patients with sarcopenia in the first line treatment, while no differences were found between sarcopenic and non-sarcopenic patients in second and third line. We acknowledge that the interpretation of these results is limited due to the small sample size and must necessarily be confirmed by subsequent larger studies.

Moreover, we cannot exclude that, in the most advanced stages of the disease, there can be a further change in body composition with a greater number of patients turning towards sarcopenia, thus confusing the basal categorization groups into sarcopenia yes/no, assessed at baseline.

To the best of our knowledge, this is the first study to evaluate sarcopenia with DXA in oncologic patients. Previous similar studies identified the CT scan as diagnostic technique. CT scans provide an estimation of three-dimensional muscle mass at the level of L3 vertebra on two-dimensional planar sections (cm^2^ of muscle tissue). The muscle area at L3 vertebra level, divided by the patient height^2^, is accepted as a marker of sarcopenia (third lumbar vertebra skeletal muscle index, L3SMI). Other studies used the measurement of the psoas muscle [28]. However, this method is certainly related to CT images which should be of high quality. The choice of the image to be analyzed is also operator dependent, requires highly-trained personnel and could vary from one center to another. Furthermore, in retrospective studies, the machines used for CT are not always the same, another point that certainly reduces the standardization of data. The use of DXA, instead, is not related with the choice of a particular skeletal level as in CT scans, allowing for better standardization of results [27]. Moreover, measuring attenuation of X-rays at two separate frequencies, it allows to obtain several indices of fat and lean mass, providing a better characterization of the real body composition. Finally, cut-off points for low muscle mass are not yet well defined for CT measurements. Regarding the measurement of psoas muscle, because it is a minor muscle, it may not be representative of overall sarcopenia [35]. ASM/heigh^2^, instead, has been shown to be one of the best predictors for estimating skeletal muscle mass and it is widely used to assess sarcopenia with precise cut-off, distinguished for gender differences [28,36].

Over time, some studies have attempted to also examine the inflammatory substrate associated with the development of sarcopenia in different clinical contexts [14,33,37,38,39,40]. For example, according to Oztürk et al. an increased NLR may have an important role in the onset of sarcopenia itself, especially in elderly people [12]. In this study, we associate body composition with inflammatory structure. According to our results, patients with sarcopenia showed significantly worse inflammatory markers compared to non-sarcopenic ones, supporting the association between sarcopenia, inflammation and worse outcomes in cancer patients treated with anti-PD-1/anti-PD-L1.

In details, patients with sarcopenia showed higher white cell counts and neutrophils, as well as higher NLR, LLR, CRP and fibrinogen. Several studies in the last years evaluated the prognostic importance of NLR and LLR as inflammatory markers in cancer patients [16,20,41]. Although different cut-off points have been used, high NLR appeared to be associated with a worse prognosis. These results has been previously reported in numerous tumors [42,43], even in patients treated with ICI [16,44]. It seems that NLR, in some extent, might reflect the patient’s degree of inflammation [42]. According to a recent meta-analysis, NLR could represent a potentially useful prognostic tool that can help in predicting survival outcome in patients treated with ICI [41]. Considering its availability and non-invasive nature, this index is particularly useful in routine clinical practice [41]. Similar considerations can be made also for LLR, another inflammatory marker able to reflect the lymphocyte cell-mediated response [45]. CRP and fibrinogen are non-specific inflammation markers; they are usually increased in oncologic patients but their significant rise in sarcopenic people can support the inflammatory substrate of sarcopenia itself. Pro-inflammatory cytokines, including TNF-α, IL-1, IL-6, are usually increased in many types of cancer [46]. In details, they represent a major player in cytokine storm, chronic inflammatory diseases, autoimmune diseases and in cancer as well. Interactions between tumour and its microenvironment through the immune system are critical for tumour development and progression. In details, IL-6 is produced in a variety of cells such as fibroblasts, endothelial cells, keratinocytes, macrophages, T cells and mast cells but also by cancer tissue and cancer cell lines. However, increased serum level of IL-6 are both due to the direct production of cancer cells and (and above all) to the monocytes’ production [47,48,49]. It can be assumed that the immune response mechanism is initiated by the tumour, but ultimately results in a paraneoplastic systemic reaction, expressed through a uniform cytokine pattern independent of the cancer form. This condition could be defined as cancer-associated dysfunctional immunostimulation, in which the inflammatory microenvironment has a significant impact in the progression of cancer [50]. In fact, higher IL-6 levels are inversely correlated with survival [51]. At the same time, sarcopenia seems to be characterized by a low-degree systemic inflammation with increased levels of pro-inflammatory cytokines. Together, these data are in line with the literature [12,15,29,32,37] which underlines the correlation between sarcopenia and the presence of a basal inflammatory status.

Accordingly, sarcopenia can reflect the increased metabolic activity of more aggressive tumors, which involves systemic inflammation and muscle wasting. As demonstrated in a cohort of 221 patients with head–neck cancer, basal sarcopenia, together with systemic inflammation (in terms of NLR), was associated with significantly lower OS and PFS [20]. In this regard, preclinical studies suggest that catabolism reduces immune response [52]. As a consequence, sarcopenia might be considered as a reliable marker of ongoing metabolic processes with consequent detrimental effect on immunotherapy efficacy. Roch et al. speculated that evolving sarcopenia is associated with a poor outcome during anti-PD-1/PD-L1 therapy by reflecting an ongoing catabolic process leading to inhibition of many features of antitumor immune response, including CD8+ T-cell migration into tumor microenvironment [34,53].

It is not possible to draw definitive conclusions, but there are several potential explanations about how body composition and sarcopenia could influence immune response and clinical outcomes during ICI treatment. The evidence that nutritional status was important for many body functions (including immune cell function) was already reported, long time before immunotherapy studies. The first 1990, a pre-clinical study showed that malnutrition impaired the ability of lymphocytes to proliferate and to product IFN-γ [54]. The presence of sarcopenia is believed to influence the host immune system leading to immune senescence [55]. Skeletal muscle cells actively modulate immune responses, both in health subjects and during illness, because they interact with immune cells as non-professional antigen presenting cells, expressing MHC-I/II and influencing T cell functions [56].

In summary, although the accurate pathophysiological mechanisms of the onset and progression of sarcopenia are complex and still not fully clarified, they could include a state of low-degree systemic inflammation characterized by an increasing level of pro-inflammatory cytokines, including TNF-α, IL-1, IL-6, and a variable level of anti-inflammatory cytokines, such as IL-10 [11,13,57,58]. As claimed by Wherry et al., the same cytokines implicated in sarcopenia, such as TGF-β and IL-6, can also be cited as mediators of T-cell exhaustion [59].

According to our results, pro-inflammatory cytokines, such as IL-6, TNF-α, TGF-α, appeared to be significantly higher in the sarcopenic group than in the non-sarcopenic one. This could confirm the hypothesis according to which chronic inflammation seems to play a role in the development of sarcopenia and might as well take part in the resistance to ICI [31,60].

When muscle atrophy is combined with elevated inflammatory cytokines (such as in cancer patients), the result is a negative impact on number and function of immune system cells.

Taken all together, these results support the hypothesis that inflammation, which is typical of sarcopenic status, through different mediators, can attenuate the efficacy of ICI [61,62]. Moreover, low skeletal muscle mass could have a considerable impact on the immune dysregulation and the efficacy of PD-1/PD-L1 inhibitors, involving poorer outcomes in cancer patients treated with these agents. Therefore, as stated, baseline body composition, and in particular skeletal muscle mass, can have a considerable impact on the efficacy of ICI, and sarcopenic status could be considered a negative predictive factor to ICI response.

The practical implication of these premises could be interesting. First, as known from KEYNOTE-024 [63], in patients with advanced NSCLC and PD-L1 expression on at least 50% of tumor cells, pembrolizumab was associated with significantly longer PFS and OS and a higher response rate. However, as can be seen from the KM curves, there is a number of patients who show rapid progression in the course of therapy in the first three months. Sarcopenia could be one of the factors responsible for this non-response, whereas its greatest impact on survival was observed right for fist-line treated patients in terms of both PFS and OS. Therefore, evaluating the sarcopenic state at baseline could orient the oncologist, in the first instance, to another therapeutic choice, such as the combined treatment pembrolizumab-chemotherapy, whose percentage of early non-responders seems to be reduced [64]. However, randomized controlled trials are definitively needed to confirm this hypothesis.

Moreover, it is important to consider that an early recognition and treatment of sarcopenia could bring benefits to patients candidates for ICI treatment. Several randomized controlled trials have demonstrated new potential treatments for cancerous cachexia such as anamorelin (a ghrelin receptor agonist) and enobosarm (a selective androgen receptor modulator), which can increase skeletal muscle mass in patients with advanced NSCLC. Exercise, supplementation with omega-3 fatty acids, melanocortin-4 receptor antagonists, inhibition of myostatin, beta-blockers, IL-6 antagonists, synthetic ghrelin and vitamin D have also been explored as possible therapies [6,65,66,67].

The main strength of this study is represented by the prospective design, as literature on this topic is only represented by retrospective studies. Moreover, the use of DXA scan instead of CT scan, certainly improves the standardization of data and adds other details on body composition.

However, the study did have limitations. First, as already has been acknowledged, our data had the important limitation of small sample size, which may limit the interpretation of the results. It should be noted that most of the studies on NSCLC patients treated with ICI share this important limitation with similar or even lower sample sizes [6,22,24,25]. Studies with higher sample size also have some limits, which lies in the inhomogeneity of the population due to multicenter data [32], different types [33] and stages of tumor [23]. The most reliable data in the literature remain those from Roch et al. [34], which include 142 patients from a single center, all with lung cancer treated both in the first and second line with anti-PD-1, with well-defined selection criteria. However, these data are retrospective as well. Nevertheless, studies with a larger cohort are needed to confirm our results. Secondly, even if a population is homogeneous for the type and stage of cancer (all patients with stage IV NSCLC), it is heterogeneous in terms of treatment, including both patients who have used PD-1/PD-L1 inhibitors as a first line and as a second and third line treatment. However, a statistical analysis was conducted separately for different lines of treatment. In conclusion, more prospective and long-term studies are certainly needed, in order to confirm inflammatory or immunological predictors of clinical outcome and therapeutic response.

## 5. Conclusions

In this proof of concept, prospective longitudinal observational study, we demonstrated for the first time using DXA scan, that NSCLC with sarcopenia at baseline frequently show worse PFS compared to subjects without sarcopenia after treatment with PD-1/PD-L1 inhibitors. Sarcopenic patients had an 8-fold higher risk of progression disease compared to non-sarcopenic patients. According to our results, sarcopenia could therefore be considered an independent unfavorable prognostic factor and predictive of a worse response to ICI. Sarcopenic impact on survival seems have even a greater impact on patients who performed PD-1/PD-L1 inhibitors as first line treatment both in terms of OS and PFS. Furthermore, sarcopenic patients showed higher levels of inflammatory biomarkers compared to non-sarcopenic patients.

However, the sample size is too low to be able to draw definitive conclusions. Studies with a larger cohort are certainly needed to confirm our results.

As a matter of fact, baseline body composition, and especially skeletal muscle mass, can have a considerable impact on the efficacy of PD-1/PD-L1 inhibitors and skeletal muscle loss might be a useful predictive marker for anti-PD-1/anti-PD-L1 therapy outcomes. Therefore, it may be important to perform a baseline assessment of sarcopenic status in NSCLC patients before starting ICI. An early diagnosis may help to identify patients more likely to achieve a better response to PD-1/PD-L1 inhibitors in routine clinical practice. Moreover, it can allow an early and targeted treatment of sarcopenia and therefore a greater efficacy ICI treatment. For certain, future perspectives of this study include larger trials aimed to demonstrate if, in first-line treatment, sarcopenia can influence survival for both treatment with ICI alone or combined with chemotherapy. Other randomized controlled trials should be performed to confirm if early treatment of sarcopenia in the patient with lung cancer candidate to initiate PD-1/PD-L1 inhibitors, is actually associated with improved survival.

## Figures and Tables

**Figure 1 cancers-13-06355-f001:**
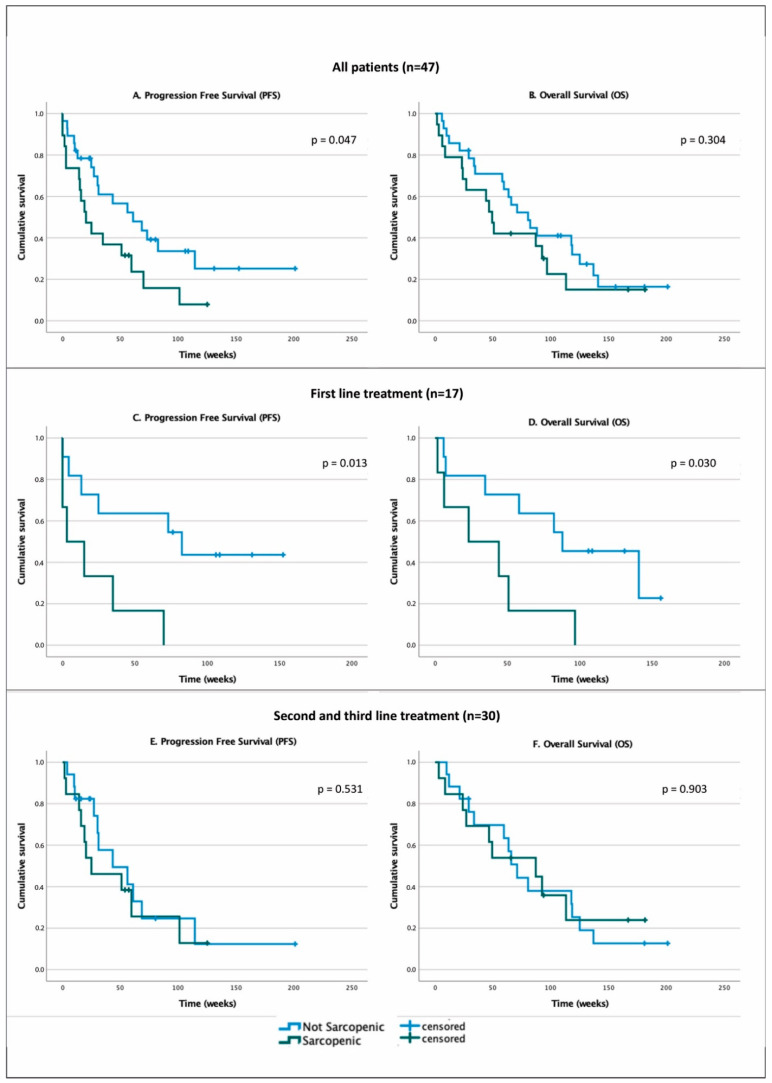
Kaplan–Meier estimates survival curves for: all patients (*n* = 47) PFS (Panel **A**) and OS (Panel **B**); first line treatment (*n* = 17) PFS (Panel **C**) and OS (Panel **D**); second and third line treatment (*n* = 30) PFS (Panel **E**) and OS (Panel **F**).

**Table 1 cancers-13-06355-t001:** Baseline characteristics of the study population. Continuous variables are expressed in median (IQR).

Total Number of Patients, N	47
Age, years, median (IQR)	67 (61; 74)
Sex	
Male, *n* (%)	27 (57.4)
Female, *n* (%)	20 (42.6)
Histotype NSCLC	
Adenocarcinoma, *n* (%)	30 (63.8)
Squamous cell carcinoma, *n* (%)	9 (19.1)
Poorly differentiated carcinoma, *n* (%)	5 (10.6)
Large cell carcinoma/mixed, *n* (%)	3 (6.4)
First line ICI, *n* (%)	18 (38.3)
Previous chemotherapy, *n* (%)	29 (61.7)
Type of ICI	
Nivolumab, *n* (%)	22 (46.8)
Pembrolizumab, *n* (%)	18 (38.3)
Atezolizumab, *n* (%)	7 (14.9)
Dead, *n* (%)	36 (76.6)
PFS, weeks (IQR)	30.3 (13.0; 73.1)
OS, weeks (IQR)	65.6 (27.0; 113.3)
ORR, *n* (%)	12 (25.5)

Abbreviations: PFS: progression free survival; OS: overall survival; ORR: overall response rate.

**Table 2 cancers-13-06355-t002:** Characteristic of metabolic and body composition parameters. Data of all patients (*n* = 47) are reported and a comparison between PD (*n* = 17) vs. CB (*n* = 30) groups. Values are expressed in median (IQR). Mann–Whitney U test was used d for continuous variables. Chi-squared was used for non-continuous variables (* *p* < 0.05).

Variables	All Patients (*n* = 47)	PD (*n* = 17)	CB (*n* = 30)	*p*
Weight, Kg	70 (58; 80)	64 (57; 82.5)	71 (57; 78)	0.782
High, m	1.70 (1.62; 1.75)	1.70 (1.62; 1.75)	1.70 (1.62; 1.74)	0.665
BMI, Kg/m^2^	23.9 (20.7; 27.8)	22.1 (20.2–28.2)	25.25 (21.87; 27.86)	0.514
BMI ≥ 25, *n* (%)	20 (42.6)	6 (35.3)	14 (46.6)	0.449
BMI 18.5–24.9, *n* (%)	26 (55.3)	11 (64.7)	15 (50)	0.330
BMI < 18.5, *n* (%)	1 (2.1)	-	1	-
Glycemia, mg/dL	97 (90; 111)	95 (88; 104)	103 (90; 116)	0.284
Insulin, mg/dL	13.2 (6.2; 17.2)	12.9 (7.9; 20.8)	13.2 (5.4; 16.8)	0.367
Total Cholesterol, mg/dL	194 (158; 234)	208 (166; 236)	189 (151; 222)	0.434
LDL Cholesterol, mg/dL	111 (78; 137)	115 (85; 141)	106 (74; 134)	0.367
HDL Cholesterol, mg/dL	50 (43; 67)	47 (43; 67)	52 (42.5; 68)	0.537
Triglycerids, mg/dL	119 (103; 168)	109 (93; 200)	133 (104; 151)	0.767
MetS				
Yes, *n* (%)	15 (31.9)	5 (29.4)	10 (33.3)	0.782
No, *n* (%)	32 (68.1)	12 (70.6)	20 (66.6)	
Lean mass total, g	46,172 (39,264; 52,784)	40,925 (36,220; 47,320)	51,864 (41,089; 53,830)	0.05 *
ASM/Ht^2^, Kg/m^2^	6.8 (5.7; 7.9)	6.3 (5.6; 7.1)	7.8 (5.9; 7.9)	0.014 *
Sarcopenia				
Yes, *n* (%)	19 (40.4)	11 (64.7)	8 (26.7)	0.011 *
No, *n* (%)	28 (59.6)	6 (35.3)	22 (73.3)	
Fat mass total, g	18,060 (14,789; 26,651)	20,574 (14,033; 30,848)	17,809 (14,781; 26,424)	0.982
Fat, %	27.5 (23.4; 34.2)	30.1 (23.4; 35.2)	27.3 (23.8; 32.5)	0.756
VAT mass, g	589 (421–955)	694 (496; 997)	570 (402; 951)	0.522

Abbreviations: ASM: appendicular skeletal muscle mass; BMI: body mass index; VAT: visceral adipose tissue.

**Table 3 cancers-13-06355-t003:** Logistic regression predicting likelihood of progression disease (PD) as best response in patients based on age, previous chemotherapy and sarcopenia.

Variables	B	SE	Wald	df	*p*	OR	95% CI for OR	
							Lower	Upper
Age	0.027	0.48	0.324	1	0.569	1.027	0.936	1.128
BMI	0.093	0.084	1.214	1	0.270	1.097	0.930	1.294
Previous chemotherapy	0.291	0.763	0.146	1	0.703	1.338	0.300	5.966
Sarcopenia	2.093	0.826	6.416	1	0.011 *	8.109	1.606	40.954
Constant	−5.602	4.078	1.887	1	0.170	0.004		

Abbreviations: BMI: body mass index. * *p* < 0.05

**Table 4 cancers-13-06355-t004:** Data on baseline full blood count, inflammatory parameters and cytokines comparing sarcopenic vs. non-sarcopenic groups. Mann-Whitney U test (* *p* < 0.05).

Variables	All Patients (*n* = 47)	Sarcopenia (*n* = 19)	No Sarcopenia (*n* = 28)	*p*
White blood cells, ×10^9^/L	7.99 (6.59; 9.25)	8.66 (7.51–11.23)	7.36 (5.9–9.1)	0.041 *
Neutrophils, ×10^9^/L	5.52 (4.38; 6.69)	6.33 (5.25–8.34)	5.01 (3.44–6.58)	0.022 *
Lymphocytes, ×10^9^/L	1.62 (1.10; 1.98)	1.51 (0.97–2.01)	1.66 (1.24–1.97)	0.305
Monocytes, ×10^9^/L	0.55 (0.38; 0.66)	0.61 (0.38–0.69)	0.50 (0.38–0.60)	0.177
Eosinophils, ×10^9^/L	0.11 (0.08; 0.27)	0.12 (0.07–0.22)	0.11 (0.08–0.30)	0.639
Basophils, ×10^9^/L	0.03 (0.02; 0.04)	0.03 (0.02–0.04)	0.03 (0.02–0.04)	0.271
NLR	3.47 (2.51; 5.16)	4.72 (2.76–6.99)	2.86 (2.08–3.90)	0.012 *
LLR	4.76 (3.8; 6.78)	6.25 (4.16–8.70)	4.39 (3.51–5.36)	0.005 *
ESR, mm/h	56 (24; 86)	59 (34–91)	43 (20–78)	0.374
CRP, µg/L	14150 (3625; 42,650)	35,200 (7300–55,900)	10,200 (3400–19,500)	0.024 *
Fibrinogen, g/L	4.91 (4.01; 6.11)	5.98 (4.10–6.38)	4.67 (3.80–5.22)	0.041 *
Ferritin, µg/L	243 (166; 394)	374 (224–452)	215 (151–326)	0.052
Transferrin, g/L	2.29 (2.04; 2.59)	2.22 (1.91–2.56)	2.30 (2.09–2.62)	0.466
IL-6, pg/mL	5.85 (3.2; 17.01)	14.3 (6.68; 22.78)	5.21 (2.80; 6.47)	0.004 *
TNF-α, pg/mL	4.34 (2.60; 5.98)	5.17 (3.49; 7.76)	3.49 (2.36; 5.17)	0.050
TGF-α, pg/mL	6.12 (3.63; 15.33)	7.68 (6.06; 10.58)	5.31 (2.74; 8.24)	0.042 *

Abbreviations: NLR: neutrophil/lymphocyte ratio; LLR: leukocyte/lymphocyte ratio.

## Data Availability

The data presented in this study are available on request from the corresponding author. The data are not publicly available due to privacy-related issues.

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
