# Peer review of "Impact of Sarcopenia and Inflammation on Patients with Advanced Non-Small Cell Lung Cancer (NCSCL) Treated with Immune Checkpoint Inhibitors (ICIs): A Prospective Study"

_cancers, 2021, doi:10.3390/cancers13246355_

Round 1

Reviewer 1 Report

Taking into account that the definition of sarcopenia implies a condition that is characterized by the loss of mass, strength and function of the muscles in older adults. Signs and symptoms include weakness, tiredness, lack of energy, balance problems, and difficulty walking and standing, and not just a biochemical or physical metric. From this point of view it would have been interesting to add these data, as well as the previous condition of the patient:
   -whether it is previously thin or not
   -at what speed has he lost weight
   -etc

Reviewer 2 Report

In a prospective study, the authors investigated whether the effects of sarcopenia and inflammation were involved in prognosis in cases of non-small cell lung cancer that received ICI.

The results and analysis are simple: prior blood sampling and other clinical parameters, muscle condition examined by X-ray absorptiometry, and, of interest, the concentration of various inflammatory cytokines, progression free survival. This is a survey of overall survival.

The results are reasonably evaluable.

But what about the association between having NSCLC in these cases and high levels of IL-6 and CRP in other carcinomas? Are there data for comparison of cytokines, etc. with and without sarcopenia before treatment for other carcinomas? At least in the literature, what about?

And then, can the condition and treatment responsiveness to ICI simply result in a difference in progression free survival? Isn't it necessary to compare cases that received treatment other than ICI with the same NSCLC (also a comparison between the sarcopeni group and the non-sarcopeni group)?

In addition, the difference in serum concentration between the presence and absence of sarcopenia, a cytokine represented by IL-6, seems to be data before treatment. But how did the ICI treatment change it? Or is it still showing a change that is related to ICI resistance?

Detailed analysis of this area is required.

Reviewer 3 Report

Authors studied the impact of sarcopenia on the efficacy of the immune checkpoint inhibitors (ICIs) in advanced non-small cell lung cancer patients. They used Dual-energy X-ray absorptiometry (DXA) and evaluated the sarcopenia of patients. Their study aim is unique and appropriate for this unsolved problem. However, this paper has several problems to prove their argument.

Major comments:

  1. As authors mentioned, the sample size is very small and conducted in single institution. This size of study cannot confirm their results.
  2. Study population consisted of exceedingly heterogenous patients. First of all, the therapeutic lines were various. Usually, the efficacy of anti-cancer drug is changed with treatment lines. This study compared the efficacy of different ICIs between different treatment lines, which can’t lead to the correct conclusion.
  3. Authors should explain the inconsistent results between PSF and OS.

Reviewer 4 Report

The manuscript is well written and the findings clearly set out. However, the English could be improved and there are several typographical errors.

Skeletal not Scheletal

The cohort number is not sufficient for the analysis to allow an accurate interpretation of the results. The conclusion that patients suffering from Sarcopenia that the ICI treatment is less effective is clear but careful 35 % (as significant number) that were not with Sarcopenia also showed PD (progression Disease).

What are or could be the practical (positive) advantages of identifying patients with Sarcopenia? Could different treatment be recommended?

The conclusions should include future perspectives for this work which obviously needs this study to be carried out on a much larger sample of patients.

Round 2

Reviewer 2 Report

Authors modified their manuscript according to the reviewers' comments. 

Reviewer 3 Report

Thank you for your revision of your paper. I found the several improvements in revised version. I am especially interested in the separate analysis of treatment lines. I expect the future studies from you. Thank you.